# Dryland Performance Tests Are Not Good Predictors of World Aquatics Points in Elite Male and Female Swimmers

**DOI:** 10.3390/sports12040104

**Published:** 2024-04-10

**Authors:** Ragul Selvamoorthy, Lewis J. Macgregor, Neil Donald, Angus M. Hunter

**Affiliations:** 1Physiology, Exercise and Nutrition Research Group, Faculty of Health Sciences and Sport, University of Stirling, Stirling FK9 4LA, UK; ragul.selvamoorthy@stir.ac.uk (R.S.); l.j.macgregor1@stir.ac.uk (L.J.M.); 2**Sport**scotland: Institute of Sport, Stirling FK9 4LA, UK; neil.donald@sisport.com; 3Department of Sport Science, Sport, Health and Performance Enhancement (SHAPE) Research Centre, Nottingham Trent University, Nottingham NG11 8NS, UK

**Keywords:** elite sport, swimming, World Aquatics, Bayesian, dryland training, jump performance, FINA points

## Abstract

*Background:* Swim performance can be reliant on strength and power. Standardisation of swim performance in different events, distances, and sexes can be completed using World Aquatics points, allowing for ranking of swimmers. The aim of this retrospective cross-sectional study was to assess whether relationships between World Aquatics points and dryland markers of performance existed in male and female elite swimmers separately and combined. *Methods:* Dryland tests included Optojump^®^ photoelectric cell countermovement jump, countermovement jump reach with a Vertec^®^ system, standing broad jump using a tape measure, repetition maximum testing in the barbell back squat, barbell deadlift, and barbell bench press. Swim performance data and dryland test data on elite male (*n* = 38) and female (*n* = 20) Scottish swimmers from 2009–2017 were collected. Swim performance data were converted to World Aquatics federation points, and Bayesian linear regression analyses examined relationships between World Aquatics points and dryland performance tests: countermovement jump height (cm) using an Optojump^®^ photoelectric cells system, countermovement jump height (cm) using a Vertec^®^ device, standing broad jump distance (cm), relative strength (load lifted (kg) per kg of body mass) in the barbell bench press (kg/kg), barbell back squat (kg/kg), barbell deadlift (kg/kg). *Results*: The Bayesian estimates of change of World Aquatics points for a unit change in jump-based measures were: Optojump^®^—men = 0.6, women = 0.6, combined = 0.4; Vertec^®^—men = 4.3, women = −1.6, combined = 2.4; standing broad jump—men = 0, women = 0, combined = 0.4. Strength-based measures were: barbell back squat—men = 2.3, women = 22, combined = −2.5; barbell deadlift—men = −5; barbell bench press—men = 41.8. *Conclusions:* Dryland performance tests are not good predictors of World Aquatics points and should rather be used for assessing training quality and monitoring injury risks.

## 1. Background

Competitive swimming is popular worldwide, with regular high-level international events occurring every year [1]. Consequently, swimming competitiveness improves annually, leading to increased investment from governing bodies and sports institutions in expertise for their athletes. A podium finish in swimming competitions can be separated by a 100 ms time difference [2], as shown in the 2020 Tokyo Olympics [3]; consequently, these podium finishes can determine the amount of funding each governing body receives per funding cycle. To maximise swim performance, practitioners and researchers seek to understand contributory factors better to inform athlete preparation. Dryland performance has been shown to contribute to overall swim performance in different events [4]. Standardising swimming performance under a single metric may allow for swimmers of various events to be grouped collectively, which can allow for relationships between swim performance and dryland performance to be explored, thereby allowing practitioners and researchers to identify contributory factors, such as lower body and upper body power and strength, to overall swim performance [4,5].

Overall swim performance may be affected by lower body power and lower and upper body strength [5]. Currently, there is no single definitive test to quantify power and strength, therefore a range of tests are widely adopted by practitioners working within high-performance sport. The most commonly adopted tests to measure lower body power in swimming and other sports include squat jumps and countermovement jumps [6,7,8] using force plates [9], contact grids [10], tape measures [11], and obstacle reaches [12]. Similarly, repetition maximum (RM) tests are the most popular to assess lower and upper body strength [13].

Dryland performance affects overall swimming performance. Therefore, practitioners might be interested in establishing relationships between dryland performance tests and swimming performance; this can help identify factors contributing to better swimming performance and that information can be used to guide training prescription. However, swim performance can be challenging to compare across different events as it is based on race time [14], increasing practitioners’ workload when assessing which factors might contribute to swim performance. Therefore, a standardised way of comparing the swim performance of varying events would be helpful. For example, sports events’ performances, classified based on time and distance, could be standardised using a common ranking system. In athletics, using World Athletics points has allowed the ranking of athletes from different events, which has also helped researchers establish relationships between lower body strength/power tests and athletic performance in field events [15]. A similar approach to the one used by World Athletics could identify relationships between standardised swim performance measured by World Aquatics (WA) points (previously known as Fédération Internationale de Natation) and dryland performance in a group of swimmers participating in different events.

Testing and monitoring for training quality are essential for improving performance and reducing injury risks, as swimmers tend to be under high training loads throughout the year and are more prone to overtraining and injuries [16]; therefore, having a routine testing monitoring schedule may allow for practitioners to prescribe loads that provide sufficient stimulus whilst mitigating overtraining and potential injury risks. However, high-performance sport is a time-sensitive environment; therefore, devising a routine schedule for testing and monitoring can sometimes be difficult, leading to inconsistent data capture and small sample sizes for data analyses. Furthermore, given the tiny margins in competitive swimming, meaningful effects may be missed due to reduced sample sizes. Null hypothesis testing is commonly used in research and applied settings; however, it can be unsuitable for small sample sizes and when the goal is to identify a clear statistical interpretation (rejecting the null hypothesis vs. the probability of something happening) [17]. In contrast, Bayesian testing can overcome such challenges, as it is based on probabilities and the models rely on prior information, thereby providing a posterior distribution that describes the uncertainty and probabilities of the data supporting in favour of or against the hypotheses [18].

Accordingly, the current study aimed to identify whether a positive relationship exists between dryland performance metrics, such as jump height/distance and relative strength, and swim performance, as measured by WA points using a Bayesian approach in males and females separately and combined. We hypothesized that there would be positive associations between jump height and WA points, and relative strength and WA points. The potential results of this study could inform testing and monitoring practices within high-performance sports for practitioners and researchers responsible for large cohorts of athletes participating in different events.

## 2. Methods

### 2.1. Study Design

This was a retrospective cross-sectional study examining the relationship between dryland performance tests and WA points using a Bayesian linear regression approach, where raw data on WA points calculated by swim times, lower body power, and lower and upper body strength were collected between 2009–2017 and were uploaded to Microsoft Excel for analysis.

### 2.2. Participants

The sample comprised of elite Scottish swimmers, which included Olympic, Commonwealth, and World Championship medallists. In order to be included in this study, swimmers had to be classed as elite, according to the criteria described below. 

There were no specific exclusion criteria, therefore, all eligible swimmers were assessed as part of this study. A total of 58 (38 male and 20 female) unique swimmers were analysed in this study, where the sample size varied between the different tests used; the highest sample size in this study was 40 (25 males and 15 females), as seen in the analysis between Vertec^®^ jump and WA points, and the lowest sample size was 9 (9 males) for the analysis between barbell deadlift and WA points. Table 1 and Table 2 provide a description of the number of swimmers, their events, and sex.

The swimmers were further classified into their respective event distances: sprint (50–100 m), middle-distance (200–400 m), and long-distance (>800 m) [19]. This cohort was considered elite if they satisfied any of the following: they were competing internationally, they were among the best swimmers in their events across Scotland, and they were funded by **sport**scotland to swim professionally [20].

The WA points for the different samples (combination of male and female, and different strokes) used are described as mean ± SD and minimum and maximum values: sprint-distance (793 ± 93, min = 415, max = 1094), middle-distance (827 ± 85, min = 496, max = 1071), long-distance (829 ± 59, min = 719, max = 925). Based on the WA point range across the three competitive swimming distances, our sample can further be considered as elite based on the work of Ruiz-Navarro et al. [21], who provided a framework of five different levels of swimming performance (>875 WA points = Tier A international competitors, >800 WA points = Tier B international competitors, >650 WA points = national-level competitors, >450 WA points = regional-level competitors, <450 WA points = recreational-level competitors).

### 2.3. Procedures

Prior to any jump testing, all participants had to undergo a dynamic warmup consisting of ten bodyweight squats, five lunges per leg, and three submaximal jumps. Whilst a traditional ramp warm up was used for strength testing.

### 2.4. Optojump®

On a stable surface, two Optojump (Microgate, Bolzano, Italy) photoelectric cells were placed parallel to each other, and the Optojump software (Version 1.10.7, Microgate, Bolzano, Italy) was used to collect jump data [8]. The participants were required to stand between the photoelectric cells in their preferred jumping stance, with their hands on their hips. Once the assessor initiated the test, the participants were asked to perform a self-selected countermovement depth and jump as high as possible while landing in the same place. A total of three attempts with thirty-second intervals between attempts were recorded, and the highest jump was taken for analysis.

### 2.5. Vertec® Jump

A standing version of the Vertec jump was used (JUMP USA, Sunnyvale, CA, USA) [12]. The Vertec testing kit was placed on a flat, stable surface, and a 30.5 cm tape was placed perpendicular to the base and directly beneath the Vertec vanes. In addition, two other 30.5 cm tapes were placed parallel on either side of the centre tape at a 45.7 cm distance; the whole area between all three lines was considered the jumpers area of action.

The participants were required to stand under the Vertec vanes, with their midfoot directly over the 30.5 cm centre tape, while keeping their toes and heels on the ground. To measure reach height, the participants were required to fully stretch their dominant hand overhead and move as many vanes as possible. These vanes were then removed from the jump test. To accommodate for varying stature, the initial setup of the Vertec base height was modulated based on the manufacturer’s guidelines. If an athlete could move two or more red vanes (15.25 cm and above), the base height was moved to the next 15.25 cm marker; this process was repeated until no more than 11 vanes were moved. Similarly, if a participant could not reach any vanes, the base height was moved down to the next 15.25 cm marker. The red vanes were spaced 15.25 cm from each other, while the blue and white vanes were spaced 3.81 cm from each other, respectively. The Vertec jump height was recorded in inches, and then converted into cm to keep units standardised.

The participants were then required to jump as high as possible with the help of an arm swing to clear as many vanes as possible. The number of vanes moved was then subtracted from the ones used for measuring reach height, and the difference was recorded as the jump height. The participants performed the jump for three trials, separated by a thirty-second interval. If a participant’s best jump height was recorded on the third trial, they were then allowed to attempt further jumps until a maximal jump was obtained or until the current maximal jump started declining.

### 2.6. Standing Broad Jump

A tape measure was placed on the floor and aligned to a straight line, with the distal end fixed to a stationary stable object. The athletes were required to stand with their toes behind the start line (line perpendicular to the beginning of the tape) and were instructed to jump as far as possible while landing in a controlled manner [11]. The measurement was taken at the nearest point of landing contact (i.e., back of the heel). A total of three attempts were made with a thirty-second interval between attempts, and the furthest distance covered in cm was recorded as jump distance.

### 2.7. Strength Tests

The standard mode of strength testing was RM testing, which included 1RM, 3RM, 5RM, and 8RM. Multiple-repetition testing has previously been shown to be a reliable method of estimating 1 repetition maximum, being more accessible to individuals who do not possess a long training history in strength training [13]. The different tests were employed for different athlete abilities, and all scores were converted to a 1RM using Landers equation [100 × repetitions × weight/(101.3 − 2.67123 × repetitions] [22].

### 2.8. Data Collection

Jump, strength, and swim performance (i.e., swim times) data for male and female swimmers between 2009–2017 were analysed. All swimmers were thoroughly familiarised with these methods prior to any testing. **sport**scotland invested in different equipment over the years; therefore, jump data were captured using three different technologies: Vertec jump (Jump USA, CA, USA), Optojump (Microgate, Bolzano, Italy), and standing broad jump. Thus, to accommodate such changes in testing procedures, data analysis involved using the years where data were available to analyse the relevant jump data (e.g., 2015–2017 Optojump). Therefore, although there are overlaps, all different technology-derived jump data were assessed separately. In addition, strength data from barbell back squat, bench press, and deadlift were normalised to body mass (taken from the highest relative strength value of the given test) and used in the analyses. All WA points were based on the base times of the corresponding years. Where base time was based on the world record time for the specific event length and stroke, the base time for the long course was agreed by the end of the year (31 December).

The data provided had missing values for both WA points and markers of performance, for example, barbell bench press and barbell deadlift data on female swimmers were not available based on our analysis timeline; therefore, we did not conduct a Bayesian linear regression for those two tests with WA points for females. Similarly, empty values were removed from the analyses. Then, implausible data points were excluded as they were deemed physiologically impossible (i.e., single-digit values for dryland tests, and double-digit values for WA points) or an outlier (i.e., in excess of 1200 WA points). Finally, any data point with WA points less than 400 were removed from the analyses, as values outside of this range are considered sub-elite or implausible. We have chosen to include WA points ranging from 400 to 1100, as regression analyses require a range of values for the statistical model to detect any associations and predictions robustly, and all data were cross-checked with competition time. Similarly, a jump height of less than 15 cm on the Optojump was removed as it was considered an anomaly for this cohort, Table 3 provides descriptive values for the dryland tests used. For multi-event swimmers, the data used in this analysis were based on the best WA point obtained during each competitive season. For example, if a swimmer had a better WA point doing a 200 m breaststroke, all jump and strength data corresponding to this performance were used in the analyses.

### 2.9. Data Analysis

All swim performance data were collected from 2009–2017, and only long-course (50 m) event data were used, as they reflected the competitive priorities of this cohort. To standardise swim performance across event length, stroke type, and sex, swim performance times were converted to WA points using the following formula.
*World Aquatics* (*P*) = 1000 × (*B*/*T*)^3^(1)
where *B* = base time, and *T* = event time of the swimmers.

The best swim performance (i.e., highest WA points) for each athlete in that given year between the months of March to August was analysed. Dryland performance tests conducted from March to June of the same year as the swimming competitions were used for analysis (mean ± SD, time between dryland test and swimming competitions = 1.8 ± 0.5 months), and the scores associated with the corresponding WA points were used for the Bayesian Linear regression. We chose a 1–3-month time window between dryland performance tests and the swimming competitions, as this was the best possible date upon which data on elite swimmers could be collected and used for analyses. The swimmers in this cohort participated in multiple events of differing distance and stroke types, which justifies the use of WA points, as it standardises performance across different strokes and distances. Therefore, the best swim performance within the competitive year was used for analysis.

### 2.10. Bayesian Analysis

A Bayesian linear regression model using stan via the rstanarm package in R was used to analyse the relationship between markers of dryland performance (jump height and relative strength) and swim performance. Stan is a probabilistic language that uses the Hamiltonian Markov chain, Monte Carlo, with the no u-turn sampler algorithm to generate posterior distributions for a given dataset.

### 2.11. Defining Models and Priors

A Cauchy distribution was assumed (approximately) for WA points, while the strength and power data were assumed to be normally distributed.

We used a Bayesian linear regression model individually for multiple predictors on WA points; the reasoning behind this was that each predictor was not interchangeable or would not be valid to include in a linear model with multiple predictors, as the methodology and technology between tests were too dissimilar [17,18].
*WA* = *β*_0_ + *β*_1_ + *ε*(2)

The intercept *β*_0_ (WA points) was given a prior mean of 650 and a standard deviation of 50. The prior mean was based on the expert suggestion of the cohort’s technical and support staff. The standard deviation represents the variability in ability levels, which allowed for both high-level and developmental swimmers to be included in the analysis.

The coefficient *β*_1_ represents the motor performance variables, which were: Optojump countermovement jump height (cm), Vertec jump height (cm), standing broad jump (cm), barbell back squat relative strength (loaded weight/body mass), barbell bench press (loaded weight/body mass), and barbell deadlift (loaded weight/body mass). ***ε*** is the residual standard deviation for each data point and reflects the linear model’s variance from the actual data points.

A prior mean of 0 and standard deviation of 5 (jump-based measures) and 0.1 (strength measures) were used for the beta coefficients. The priors used were based on expert knowledge from technical and support staff on the expected change in WA points based on a unit change in the predictors.

The Bayesian analyses conducted generate Bayesian estimates of size with 95% high-density intervals, which can be considered equivalent to beta coefficients and 95% confidence intervals generated in frequentist linear regressions, for example for an assumed estimate of size of 15 of a dependent variable would imply that for every 1 unit increase in the variable, there will be a 15 unit increase in World Aquatics point.

## 3. Results

There were no plausible relationships between jump and strength performance on WA points in both male and female swimmers, combined and separately, as analysed through the Bayesian linear models. Table 4 and Table 5 describe the estimates of size and 95% high-density intervals (HDI) for the Bayesian models.

### 3.1. Relationship between Jump Height and WA Points

The Bayesian estimates of the change in WA points due to a unit increase in the markers of jump performance for men were: Optojump jump height = 0.6 (95% HDI: −2.8–3.8), Vertec jump reach height = 4.3 (95% HDI: 1.9–6.7), standing broad jump = 0 (95% HDI: −0.2–0.3). For women: Optojump jump height = 0.6 (95% HDI: −4–5.1), Vertec jump reach height = −1.6 (95% HDI: −4.2–1.6), standing broad jump = 0 (95% HDI: −0.1–0.2). For men and women combined data: Optojump height = 0.4 (95% HDI: −2.2–2.9), Vertec jump reach height = 2.4 (95% HDI: 0.8–4.1), standing broad jump = 0.1 (95% HDI: −0.1–0.2).

### 3.2. Relationship between Strength Measures and WA Points

The linear regression estimates of the Bayesian model for a unit change of strength performance (measured in kg per kg body mass [kg/kg]) on WA points in men were barbell back squat = 2.3 (95% HDI: −38.8–80.8), barbell deadlift = −5 (95% HDI: −60–50.8), barbell bench press = 41.8 (95% HDI: −155.5–230). In women, the estimates were: barbell back squat = 22 (95% HDI: −257.7–304.3). For men and women combined data: barbell back squat = −2.5 (95% HDI: −78.1–74.3).

## 4. Discussion

We aimed to identify relationships between dryland power and strength and swim performance standardised by WA points. We used a Bayesian regression analysis to assess statistically small but potentially meaningful effects in a cohort of elite swimmers, a population that is challenging to research [19,23]. The Bayesian estimates of the size and posterior density plots indicate no plausible relationships between any jump or strength performance variables on WA points in males and females, separately and combined, with too many uncertainties in the data, as shown in the high-density intervals; these results may be considered contradictory to the vast pool of research that has examined the relationship between dryland performance and swimming performance [4,7,9,24]; however, most of those studies have only looked at the relationship between dryland performance tests and swimming performance as a measure of time, within cohorts of swimmers of the same event, thereby reducing the effectiveness of analysing swimmers of varying events. In contrast, our study used WA points to standardise swimming points across multiple swimming events, allowing relationships between dryland performance tests and WA points to be explored in swimmers competing in various events, which may be the reason we found no relationships. The lack of relationship between jump and lower and upper body strength performance variables on standardised swim performance contradicts our initial hypothesis, where we expected to see a positive relationship between dryland and swim performance.

Our study is the first to examine the relationships between different dryland performance tests and standardised swim performance in a cohort of elite male and female swimmers with varying lengths of event and strokes, separately and combined. A similar type of analysis has previously been performed on national-level swimmers, where Garrido et al. [24] clustered swimmers of different strokes competing in 100 m and 200 m distance events; the authors found a positive relationship between isometric handgrip strength and WA points in juvenile males competing in 100 m events and female swimmers of all ages competing in both 100 m and 200 m events. Similarly, in athletics, Aoki et al. [15] have shown an association between greater lower body power and strength and improved standardised athletic performance (i.e., World Athletics points) in a mix of sprint, jump, and throwing athletes. The difference between our work and those of Garrido et al. and Aoki et al. lies in the fact that their samples, although diverse, were from events that required high levels of strength and power as part of their sport, whilst in swimming, the longer the event distance, the less this becomes a requisite. The cohort sample used in this study was diverse, as we had swimmers of all stroke types and distances. Using WA points enhances the decision-making of researchers and practitioners by standardising the swim performance of different events, allowing regression analyses to be conducted between swim performance and dryland performance.

There was very little to negligible association between WA points and any dryland performance tests in this study, indicating that the dryland tests used in this study are not good predictors of standardised swim performance. The tests we used only reflect one aspect of multi-dimensional swimming performance. We attempted to use WA points as an overall marker of swim performance, which contrasts with other studies that have only looked at relationships between specific swimming event performance times and dryland performance [4,7,9,24], which can be inefficient for coaches who may work with swimmers competing in various events. The coaches of this cohort of swimmers (Tigg and Wright, personal communication, November 2022) explained that the results of this study could be attributed to reduced shoulder rotational range of motion, reliance on improper muscle groups for technique execution, and longer upper limb anthropometry. We did not have the abovementioned variables in our study, which could be confounding variables affecting our results, as these variables could affect stroking patterns [25,26], affecting overall swim performance, whilst such an effect would not be present in bench press performance or the barbell squat and deadlift [27,28,29], due to the unassessed variables discussed above. This study was intended to be an initial observational work. Additionally, the dryland tests used in the Bayesian linear regression with WA points were conducted 1–3 months before the swimming competitions, where some swimmers were in different training phases, such as the special preparatory phase and competitive phase, which could also have affected the results that we have obtained, as some swimmers might have achieved greater physical performance peak as a result of tapering, while others may have still been focused on developing physical qualities [30]. Therefore, we recommend that future studies incorporate more dryland and in-water performance measurements to identify the causative factors of the relationship between dryland performance and standardised swim performance.

We performed a Bayesian regression analysis between WA points and dryland performance metrics of men and women, combined and separately. We did not find any differences in the relationships observed in either combined or individual sex data. These findings are important, as females are often underrepresented in sports performance research due to perceived confounding variables such as strength differences, motor performance, and menstrual cycle phase [31,32,33,34]. The lack of relationships between dryland performance and WA points in both sexes adds further support to the growing body of work emphasising that female athletes should not be excluded from sports performance research [35].

Elite swimming involves tight margins, such as a podium finish that can be separated by a 100 ms time difference [2]; therefore, identifying small effect sizes (Cohen’s D > 0.2, relative ratio > 1.05) can become necessary when regression analyses are conducted [18]. The Bayesian analysis incorporates prior information based on expert knowledge in the statistical model. It, therefore, reduces the requirement for very large sample sizes to identify these small effects, whereas, in frequentist statistics, an a priori sample size calculation would have been required [18].

Our study had samples as high as 40 (25 male, 15 female) and as low as 9 (9 male), as different dryland tests had varying samples. Small sample sizes could be interpreted as a limitation; however, our sample sizes reflect the norm seen in elite swimming research [23,24]. Access to elite swimmers is difficult and rare due to competition and training schedule demands, which makes our study important for coaches and researchers who are interested in dryland factors that may affect swim performance and are in the process of selecting assessment methods for their squads.

## 5. Conclusions

Despite the value of WA points for ranking swimmers as a standardised metric for swim performance, we are the first to demonstrate no relationships between power and strength measures captured on dryland and WA points in an elite cohort of swimmers of different event lengths and stroke types. The findings of this study are important as they highlight the need for capturing measurements of technical performance, anthropometry, and biomechanical data when researchers and practitioners aim to identify relationships between non-swim tests and WA points. A Bayesian statistical approach provides an important avenue for researchers and practitioners who may struggle with sample sizes, as they may use data from previous research and/or practitioner experience.

## Figures and Tables

**Table 1 sports-12-00104-t001:** Sample distribution between sexes, event distance, and strokes for the jump-based measure analyses with WA points.

*Optojump*							Total
Sex	Distance	Freestyle	Breaststroke	Backstroke	Butterfly	Medley	*n*
		(*n*)	(*n*)	(*n*)	(*n*)	(*n*)	
	Sprint	4	2	1			
Male	Middle	3	3		1	2	
	Long	1					
Total		8	5	1	1	2	17
	Sprint		2	1			
Female	Middle					2	
	Long	1					
Total		1	2	1		2	6
*Vertec*							Total
Sex	Distance	Freestyle	Breaststroke	Backstroke	Butterfly	Medley	*n*
		(*n*)	(*n*)	(*n*)	(*n*)	(*n*)	
	Sprint	2	8	3			
Male	Middle	2	4		2	4	
	Long						
Total		4	12	3	2	4	25
	Sprint	3	1	2	2		
Female	Middle	4	1	1			
	Long	1					
Total		8	2	3	2		15
*Standing broad jump (SBJ)*							Total
Sex	Distance	Freestyle	Breaststroke	Backstroke	Butterfly	Medley	*n*
		(*n*)	(*n*)	(*n*)	(*n*)	(*n*)	
	Sprint	4	5	1			
Male	Middle	3	2		1	3	
	Long						
Total		7	7	1	1	3	19
	Sprint	2		2	2		
Female	Middle	3	1				
	Long	1					
Total		6	1	2	2		11

**Table 2 sports-12-00104-t002:** Sample distribution between sexes, event distances, and strokes for strength-based test analyses with WA points.

*Barbell back squat*							Total
Sex	distance	Freestyle	Breaststroke	Backstroke	Butterfly	Medley	(*n*)
		(*n*)	(*n*)	(*n*)	(*n*)	(*n*)	
	Sprint	5	7	4			
Male	Middle	4	5	1	1	2	
	Long	1					
Total		10	12	5	1	2	30
	Sprint	1					
Female	Middle	1					
	Long						
Total		2					2
*Barbell deadlift*							Total
Sex	distance	Freestyle	Breaststroke	Backstroke	Butterfly	Medley	(*n*)
		(*n*)	(*n*)	(*n*)	(*n*)	(*n*)	
	Sprint	1	2	1			
Male	Middle	2	2			1	
	Long						
Total		3	4	1		1	9
	Sprint						
Female	Middle						
	Long						
Total							
*Barbell bench press*							Total
Sex	distance	Freestyle	Breaststroke	Backstroke	Butterfly	Medley	(*n*)
		(*n*)	(*n*)	(*n*)	(*n*)	(*n*)	
	Sprint	4	5	2	1		
Male	Middle	4	1			1	
	Long	1					
Total		9	6	2	1	1	19
	Sprint						
Female	Middle						
	Long						
Total							

**Table 3 sports-12-00104-t003:** Dryland performance test descriptives for male and female elite swimmers.

	Male			Female		
	Mean ± SD	Min	Max	Mean ± SD	Min	Max
Optojump^®^(cm)	45.4 ± 5.56	37.1	55.2	33.52 ± 4	30.4	42.8
Vertec^®^(cm)	61.81 ± 8.37	45.7	88	46.64 ± 8	35.6	67.5
SBJ (cm)	260.83 ± 17.54	231	289	204.15 ± 15.91	178	231
Back Squat (kg/kg)	1.41 ± 0.21	1.06	1.88	1.5 ± 0.03	1.48	1.52
Bench Press (kg/kg)	1.26 ± 0.16	0.8	1.72			
Deadlift (kg/kg)	1.8 ± 0.31	1.42	2.31			

SBJ = Standing broad jump.

**Table 4 sports-12-00104-t004:** Bayesian estimates and high-density intervals of the linear relationship between jump-based measures and WA points.

	Male		Female		Combined	
*Optojumpcmj*	*Estimates*	*HDI*	*Estimates*	*HDI*	*Estimates*	*HDI*
Intercept (WA)	803.2	668.6–944.2	820.8	679.8–961.2	813.1	710.1–909.7
Jump height (cm)	0.6	-2.8–3.8	0.6	−4–5.1	0.4	−2.2–2.9
*Vertecmj*	*Estimates*	*HDI*	*Estimates*	*HDI*	*Estimates*	*HDI*
Intercept (WA)	561.9	411–710.2	846	722.8–970	671.6	577.5–762.7
Jump reach height (cm)	4.3	1.9–6.7	−1.6	−4.2–1.6	2.4	0.8–4.1
*Standing broad jump (cm)*	*Estimates*	*HDI*	*Estimates*	*HDI*	*Estimates*	*HDI*
Intercept (WA)	829.3	752.2–909.4	787.3	742.6–834.3	804.8	758.1–852.6
Jump distance (cm)	0	−0.2–0.3	0	−0.1–0.2	0.1	−0.1–0.2

**Table 5 sports-12-00104-t005:** Bayesian estimates and high-density intervals of the linear relationship between relative strength in the barbell back squat, bench press, and deadlift, and WA points.

	Male		Female		Combined	
*Back squat*	*Estimates*	*HDI*	*Estimates*	*HDI*	*Estimates*	*HDI*
Intercept (WA)	838.1	727.7–954.5	679.2	242–1106.5	840	725.8–952.2
Relative strength(kg/kg)	2.3	−38.8–80.8	22	−257.7–304.3	−2.5	−78.1–74.3
*Deadlift*	*Estimates*	*HDI*				
Intercept (WA)	853.4	716.6–982.1				
Relative strength (kg/kg)	−5	−60–50.8				
*Bench press*	*Estimates*	*HDI*				
Intercept (WA)	801.8	566–1045				
Relative strength (kg/kg)	41.8	−155.5–230				

Deadlift and bench press data were only available in males, and not females.

## Data Availability

The data that support the findings of this study are available from the head of performance solutions at the **sport**scotland Institute of Sport, but restrictions apply to the availability of these data, which were used under license for the current study, and so are not publicly available. Data are, however, available from the authors upon reasonable request and with permission of the Head of Performance Solutions at the **sport**scotland Institute of Sport.

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
