# Peer review of "Dryland Performance Tests Are Not Good Predictors of World Aquatics Points in Elite Male and Female Swimmers"

_sports, 2024, doi:10.3390/sports12040104_

Round 1

Reviewer 1 Report (Previous Reviewer 1)

Comments and Suggestions for Authors

Dear authors,

Congratulations on making the suggested changes to improve your manuscript.

After reviewing the responses to the comments, the type of study conducted still does not appear in the study abstract. Again, I suggest that you amend this and add the type of studies conducted in this section.

Kind regards.

Author Response

Dear Reviewer,

We thank you for your valuable feedback. We believe that including "a retrospective cross-sectional study" in line 13 would have indicated the type of study we used. If the reviewer did not mention this, we would appreciate it if they could give us more details on how we could address their comment.

Kind regards

Reviewer 2 Report (Previous Reviewer 3)

Comments and Suggestions for Authors

The authors of the paper titled 'Dryland performance tests are not good predictors of World Aquatics Points in elite male and female swimmers' are acknowledged for their appropriate responses and corrections.

Author Response

Dear reviewer,

We thank you for your time and consideration in providing feedback to our manuscript.

Kind regards,
Ragul

Reviewer 3 Report (New Reviewer)

Comments and Suggestions for Authors

General comment:

Note in your text are many editorial errors. Capitol letter in the middle of the sentence, red letters, words in bold...

Abstract

Provide a sample size of your study.

What mean kg/kg? I suppose it is mean ratio between body mass and load lifted? It must be explained.

Line 25-26 here is unnecessary break. Also, in the results briefly describe how to interpret provided values.

Introduction

Generally, introduction is well written but:

Line 50 – provide more rationality of used performance test for swimming instead of list what test, and what devices were used.

Last paragraph- provide the practical application of potential results.

M & M

Participants – inclusion – exclusion criteria are unclear. Please explain in more detail.

Table 3 should be moved into results section.

Is table 5 missing with data?

Discussion

Provide limitations and strength of your study.

Author Response

General comment:

Note in your text are many editorial errors. Capitol letter in the middle of the sentence, red letters, words in bold...

We thank the reviewer for their feedback, we have carefully edited the text, and now included red letters to emphasise amendments made based on reviewer comments. The words in bold such as sportscotland is a reflection of the governing body’s requirement on emphasising sport in bold. And the use of capital letters in the middle of a sentence is to emphasise either location, equipment, methodology names.

Abstract

Provide a sample size of your study.

This is a valuable comment that we have now added (Male = 38, Female = 20) to the updated manuscript in line 19.

What mean kg/kg? I suppose it is mean ratio between body mass and load lifted? It must be explained.

We acknowledge the reviewer’s concern, you are correct kg/kg indicates load lifted/kg of body mass and have now included the following (load lifted (kg) per kg of bodymass) in line 24 of the abstract.

Line 25-26 here is unnecessary break. Also, in the results briefly describe how to interpret provided values.

We are thankful for the reviewer’s eye for detail, this has now been corrected. We have provided an indication on how to interpret the provided values with the following “The Bayesian estimates of change of World Aquatics points for a unit change in jump-based measures” in lines 26-27. As this is a linear regression, we have mentioned that for a unit change in any of the dryland test measures, we should see an “x” amount of increase in WA points. In order to keep to the abstract word limit, we have limited the amount of information, however in the main body text we have expanded upon this point, and ensured a clear description has been provided.

Introduction

Generally, introduction is well written but:

Line 50 – provide more rationality of used performance test for swimming instead of list what test, and what devices were used.

Dear reviewer, we thank you for your valuable insight, we have provided the rationale of the performance test with the following “Overall swim performance may be affected by lower body power, lower body, and upper body strength [5]. Currently, there is no single definitive test to quantify power and strength, therefore a range of tests are widely adopted by practitioners working within high-performance sport. Among the most commonly adopted tests to measure lower body power in swimming and other sports include squat jumps and counter-movement jumps [6-8] using force plates [9], contact grids [10], tape measures [11], and obstacle reaches [12]. Similarly, repetition maximum (RM) tests are the most popular to assess lower and upper body strength [13].” in lines 51-58.

Last paragraph- provide the practical application of potential results.

We appreciate the reviewer’s valuable comment, and have now added the following “The potential results of this study could inform testing and monitoring practices with-in high performance sports, for practitioners and researchers responsible for large co-horts of athletes participating in different events.” In lines 95-98 of the updated manuscript.

M & M

Participants – inclusion – exclusion criteria are unclear. Please explain in more detail.

We acknowledge the reviewer’s concern, and have provided the following “In order to be included in this study, swimmers had to be classed as elite, according to the criteria described below, there were no specific exclusion criteria, therefore all eligible swimmers were assessed as part of this study”, furthermore, a description of the types of swimmers used in this study, and how they were classified as elite is provided in lines 120-133.

Table 3 should be moved into results section.

We thank the reviewer for their feedback, we would like to mention that table – 3 is a table of descriptive values consisting of mean, standard deviation, minimum and maximum value to provide context on the level of physical abilities that the swimmers of this study had, and as such we feel this would fit better in the methods section. However, we are happy during the copy editing process for this table to be relocated if the editor feels the table could be placed in a better position.

Is table 5 missing with data?

Dear reviewer, we have provided an explanation in lines 203-205 that due to our strict testing timelines for data analyses, female data on bench press and deadlift were not available for analyses, hence the blank data columns in the table. To avoid confusion, we have reformatted the table, and added the following footnote “Deadlift and bench press data were only available in Males, and not Females”.

Discussion

Provide limitations and strength of your study.

Dear reviewer we have provided the limitations and strengths of our study in the lines 374-387, as we explain that our study had small sample sizes, and the use of Bayesian analyses reduces the limitations of such small sample sizes. If the reviewer thinks it Is necessary a distinct limitations sub-section we could reformat the text to address their concern.

This manuscript is a resubmission of an earlier submission. The following is a list of the peer review reports and author responses from that submission.

Round 1

Reviewer 1 Report

Comments and Suggestions for Authors

Abstract:

Add the name of the methods in this section

Add the type of study carried out (descriptive, pilot, review, cross-sectional, etc.).

Specify the methodology section (it is recommended to consult the journal's instructions).

Background:

Line 36: Improve the wording of this sentence and use a linguistic connector to make sense of what you are putting in.

A difference of < 0.01s in final time in a swimming competition can separate podium and non-podium finishes [2], as was shown in the recent 2020 Tokyo…

Line 45: Examples of these factors and references to scientific literature

…contributory factors better to inform athlete preparation.

Line 53: Relate to the previous paragraph using a linguistic connector

Dryland performance affects overall swimming performance; practitioners might be…

Methods:

Specify the type of study you are undertaking and justify it in a subsection entitled 'Study design'. This section is essential for understanding the study you are conducting.

Participants:

Add up the total number of swimmers

It is recommended to include literature that has used the tests proposed in the study.

Discussion:

Line 275-283: Contrast your statement with the scientific literature and a strong opinion.

Lines 302-317: Contrast your statement with the scientific literature and a strong opinion.

Lines 318-326: Good for the claims in this paragraph. But more studies can be referenced than talking about the paucity of literature comparing the differences between men and women and their exclusion from studies.

Lines 327-328: Add some figures or an approximation, even for studies with small samples.

Elite swimming involves tight margins and identifying small effect sizes can become necessary when regression analyses are conducted.

It would be good to add this in the methodology section...

Our study had samples as high as 40 (21 males, 19 females) and as low as 6 (5 males, 332

1 female)

References:

Please read this section in its entirety, taking into account the rules of the journal. Check that the journal names are in italics.

Author Response

Abstract:

Add the name of the methods in this section

We thank the reviewer for taking the time to provide such valuable feedback on the manuscript. We have amended the manuscript as suggested and hope that these amendments have addressed the reviewer’s concerns. We have now added the name of the methods used in the methodology sub-section of the abstract with the following lines “Dryland tests included optojump photoelectric cell countermovement jump, countermovement jump reach with a Vertec system, standing broad jump using a tape measure, repetition maximum testing in the barbell back squat, barbell deadlift, and barbell bench press” in lines 15-18 of the updated manuscript.

Add the type of study carried out (descriptive, pilot, review, cross-sectional, etc.).

Specify the methodology section (it is recommended to consult the journal's instructions).

Thank you for pointing out this omission, we have now indicated that this is a retrospective cross-sectional study in line 13 of the updated manuscript. We have now included the methodology section within the abstract in line 15 of the updated manuscript.

Background:

Line 36: Improve the wording of this sentence and use a linguistic connector to make sense of what you are putting in.

A difference of < 0.01s in final time in a swimming competition can separate podium and non-podium finishes [2], as was shown in the recent 2020 Tokyo…

Thank you for highlighting this lack of clarity, we have reworded the sentence as follows “A podium finish in swimming competitions can be separated by a 100 ms time difference [2], as shown in the 2020 Tokyo Olympics [3]; consequently, these podium finishes can determine the amount of funding each governing body receives per funding cycle.” In lines 39-42 of the updated manuscript.

Line 45: Examples of these factors and references to scientific literature

…contributory factors better to inform athlete preparation.

We thank the reviewer for providing the opportunity to include examples of what factors may contribute to enhanced WA points. We have now added the following line “to identify contributory factors, such as lower body and upper body power and strength, to overall swim performance [4-5]” in lines 48-49 of the updated manuscript.

Line 53: Relate to the previous paragraph using a linguistic connector

Dryland performance affects overall swimming performance; practitioners might be…

Thank you for your suggestion, we have added the linguistic connector “therefore” in line 56, after the end of the first sentence terminating with the word “performance”.

Methods:

Specify the type of study you are undertaking and justify it in a subsection entitled 'Study design'. This section is essential for understanding the study you are conducting.

We thank the reviewer for their constructive comment, we have now added a “study design” section, along with the following  This was a retrospective cross-sectional study examining the relationship Between dryland performance tests and WA points using a Bayesian linear regression approach, where raw data on WA points calculated by swim times, lower body power, and lower and upper body strength were collected between 2009-2017 and were uploaded to Microsoft Excel for analysis.” In lines 94-99 in the updated manuscript.

Participants:

Add up the total number of swimmers

We agree with the  reviewer that the total number of swimmers is an important piece of methodological detail, we have now provided the total number of swimmers which is 58 (30 Male, 28 Female) in lines 102-103 of the updated manuscript.

It is recommended to include literature that has used the tests proposed in the study.

Thank you for this recommendation, we have now included references to literature that highlights the validity and reliability of the optojump, vertec, and standing broad jump tests, in line 134, line 141, and line 169 of the updated manuscript. We have already provided references for the strength and Bayesian analyses, therefore we did not add any new references.

Discussion:

Line 275-283: Contrast your statement with the scientific literature and a strong opinion.

Lines 302-317: Contrast your statement with the scientific literature and a strong opinion.

We thank the reviewer for this insightful feedback, we have now included contrasting statements with the scientific literature and strong opinions by adding the following “these results may be considered contradictory to the vast literature of research that has examined the relationship between dryland performance and swimming performance [4,7,9,25]; however, most of those studies have only looked at the relationship between dryland performance tests and swimming performance as a measure of time, within cohorts of swimmers of the same event, thereby reducing the effectiveness of analysing swimmers of varying events” in lines 293-299 of the updated manuscript, and  “We attempted to use WA points as an overall marker of swim performance, which contrasts with other studies that have only looked at relationships between specific swimming event performance times and dryland performance [4,7,9,25], which can be inefficient for coaches who may work with swimmers competing in various events.” In lines 327-331 of the updated manuscript.

Lines 318-326: Good for the claims in this paragraph. But more studies can be referenced than talking about the paucity of literature comparing the differences between men and women and their exclusion from studies.

Thank you for this suggestion, we agree that inclusion of further references to highlight the paucity of literature in this area will enhance our discussion. We have now added 5 references [31-35] in total to support our claims in line 348 of the updated manuscript.

  1. McNulty KL, Elliott-Sale KJ, Dolan E, Swinton PA, Ansdell P, Goodall S, Thomas K, Hicks KM. The effects of menstrual cycle phase on exercise performance in eumenorrheic women: a systematic review and meta-analysis. Sports Medicine. 2020;50(10):1813-1827.
  2. Elliott-Sale, KJ, Minahan, CL, de Jonge XAJ, Ackerman, KE Sipilä, S, Constantini NW, Lebrun CM Hackney AC. Methodological considerations for studies in sport and exercise science with women as participants: a working guide for standards of practice for research on women. Sports Medicine. 2021;51(5):843-861.
  3. Navalta JW, Davis DW, Stone WJ. Implications for cisgender female underrepresentation, small sample sizes, and misgendering in sport and exercise science research. Plos one. 2023;18(11), p.e0291526.
  4. McNulty K, Olenick A, Moore S, Cowley E. Invisibility of female participants in midlife and beyond in sport and exercise science research: a call to action. British Journal of Sports Medicine. 2024;58(4):180-181.
  5. Costello JT, Bieuzen F, Bleakley CM. Where are all the female participants in Sports and Exercise Medicine research? Eur J Sport Sci. 2014;14(8):847–51.

Lines 327-328: Add some figures or an approximation, even for studies with small samples.

Elite swimming involves tight margins and identifying small effect sizes can become necessary when regression analyses are conducted.

We thank the reviewer for their comment, we have indicated “a time difference of less than 100 ms” and “Cohen’s D > 0.2, relative ratio > 1.1” in the text in lines 353-354 of the updated manuscript.

It would be good to add this in the methodology section...

Our study had samples as high as 40 (21 males, 19 females) and as low as 6 (5 males, 1 female)

Thank you for this suggestion, we have added this information in the methodology section to improve the clarity of this communication: “the highest sample size in this study was 30 (11 males and 19 females) as seen in the analysis between barbell back squat and WA points, and the lowest sample size was 9 (2 males and 7 females) for the analysis between barbell deadlift and WA points in lines 104-107 with the amended values, which are also reflected in the discussion “Our study had samples as high as 30 (11 male, 19 female) and as low as 9 (2 male, 7 female)” in lines 359-360 of the updated manuscript.

References:

Please read this section in its entirety, taking into account the rules of the journal. Check that the journal names are in italics.

We thank the reviewer for their time in verifying the formatting of our references, we have now amended the references, ensuring that journal names are in italics.

Reviewer 2 Report

Comments and Suggestions for Authors

This is an interesting study on whether swimming performance measured with WA points may be predicted via lower body power and whole body strength. Authors have done an 8 years research and I would like to comment their efforts. Although the manuscript is well written and well organized, there seems to be a fatal problem with the methodology used.

Normally, when a regression or correlation analysis is used, the variables are all taken together in the same time of the training cycle. That doesn't seem to be the case here. Authors used the best annual swimming performance (lines 205-206) and perhaps randomly correlated it with measures of power and strength. No details are presented for this matter inside the manuscript.

Also, no details are presented regarding the time-points where power and strength were measured. For example, strength and power might be measured during the general preparation phase or during the pre-competition phase, while best swimming performance might be measured during a competition 4-5 weeks away from power and strength measurements.   

Giving the fact that this is a correlational study, there should be a consistency between the length of time that mediates between the variables. 

Author Response

This is an interesting study on whether swimming performance measured with WA points may be predicted via lower body power and whole body strength. Authors have done an 8 years research and I would like to comment their efforts. Although the manuscript is well written and well organized, there seems to be a fatal problem with the methodology used.

We thank the reviewer for their valuable input and description of our study. We acknowledge their concern regarding timing of dryland performance tests, and subsequent swimming performance data that was collected, in response to these concerns and through addressing comments provided by the other reviewers, we have amended the manuscript to address the reviewer’s concerns.

Normally, when a regression or correlation analysis is used, the variables are all taken together in the same time of the training cycle. That doesn't seem to be the case here. Authors used the best annual swimming performance (lines 205-206) and perhaps randomly correlated it with measures of power and strength. No details are presented for this matter inside the manuscript.

Also, no details are presented regarding the time-points where power and strength were measured. For example, strength and power might be measured during the general preparation phase or during the pre-competition phase, while best swimming performance might be measured during a competition 4-5 weeks away from power and strength measurements.  

Giving the fact that this is a correlational study, there should be a consistency between the length of time that mediates between the variables.

We thank the reviewer for their insightful feedback in raising these points. We apologise for this confusion. We have updated the methods section with the following lines “The most recent dryland performance test score associated with the highest WA point, dated no later than three months before the competition, was used for the Bayesian Linear regression, where most swimmers were in the competition phase of their training regime” in lines 216-219 of the updated manuscript. Although, we have a few months between the dryland tests and the swimming competition, they were conducted during the same training phase. We hope that this clarification within the manuscript serves to resolve the reviewer’s concerns regarding data collection and analysis.

Reviewer 3 Report

Comments and Suggestions for Authors

The research explored the relationship between dryland performance metrics and World Aquatics points within the Scottish elite swimmers. The results highlighted a lack of substantial correlations between dryland metrics and World Aquatics points, indicating their limited effectiveness in predicting swimming performance rankings. These findings emphasise the need to shift the focus of dryland assessments towards assessing training effectiveness and recognising injury susceptibility in elite swimmers.

Comments and Suggestions for Authors

Please format the title text considering the journal template and journal instructions for authors

Please correct the lines in Tables 3 & 4

Finally, congratulations to the authors for correlating strength and swimming performance standardized by WA points. However, the tests mainly focused on lower body strength, while it is well known that swimming performance, except for the breaststroke, primarily relies on arm action. To improve the study's clarity, it may be beneficial to remove the barbell bench press test and consider changing the title.

I believe that this observation can be accepted by the authors according to their reference to lines 314-317.

Author Response

The research explored the relationship between dryland performance metrics and World Aquatics points within the Scottish elite swimmers. The results highlighted a lack of substantial correlations between dryland metrics and World Aquatics points, indicating their limited effectiveness in predicting swimming performance rankings. These findings emphasise the need to shift the focus of dryland assessments towards assessing training effectiveness and recognising injury susceptibility in elite swimmers.

We thank the reviewer for their positive feedback and observations regarding our study and its impact to the field.

Comments and Suggestions for Authors

Please format the title text considering the journal template and journal instructions for authors

Thank you for highlighting these formatting issues. We have now formatted the title text to reflect the journal template and journal instructions for authors.

Please correct the lines in Tables 3 & 4

We thank the reviewer for highlighting these anomalies, we have increased the line space in tables 3 & 4, and ensured the formatting is consistent.

Finally, congratulations to the authors for correlating strength and swimming performance standardized by WA points. However, the tests mainly focused on lower body strength, while it is well known that swimming performance, except for the breaststroke, primarily relies on arm action. To improve the study's clarity, it may be beneficial to remove the barbell bench press test and consider changing the title.

I believe that this observation can be accepted by the authors according to their reference to lines 314-317.

We thank the reviewers for their recommendation. Although, a very valuable suggestion, we believe that including barbell bench press is important, as many practitioners use it in their practise as part of their testing and training protocols with their swimmers. Having an insight on the impact of barbell bench press performance on overall swim performance can aid the practitioners to make better informed decisions regarding training and testing, and it is therefore important to include these data within our manuscript.

Round 2

Reviewer 2 Report

Comments and Suggestions for Authors

Dear Authors,

I fully understand that the current research is a it is a huge research project with many difficulties. I also appreciate the fact that the participants of the study were well-trained swimmers. However, correlational analysis should include variables which were measured close to each other.